# AuTAgent: A Reinforcement Learning Framework for Tool-Augmented Audio Reasoning

**Siqian Tong** [1 2]  **Xuan Li** [1 2]  **Yiwei Wang** [3]  **Baolong Bi** [2 4]  **Yujun Cai** [5]
**Shenghua Liu** [2 4]  **Yuchen He** [2 4]  **Chengpeng Hao** [1 2]

tongsiqian.github.io/AuTAgent

## Abstract

Large Audio Language Models (LALMs) excel at perception but struggle with complex reasoning requiring precise acoustic measurements. While external tools can extract fine-grained features like exact tempo or pitch, effective integration remains challenging: naively using all tools causes information overload, while prompt-based selection fails to assess context-dependent utility. To address this, we propose **AuTAgent** (**Au**dio **T**ool **Agent**), a reinforcement learning framework that learns when and which tools to invoke. By employing a sparse-feedback training strategy with a novel Differential Reward mechanism, the agent learns to filter out irrelevant tools and invokes external assistance only when it yields a net performance gain over the base model. Experimental results confirm that AuTAgent complements the representation bottleneck of LALMs by providing verifiable acoustic evidence. It improves accuracy by 4.20% / 6.20% and 9.80% / 8.00% for open-source and closed-source backbones on the MMAU Test-mini and the MMAR benchmarks, respectively. In addition, further experiments demonstrate exceptional transferability. We highlight the complementary role of external tools in augmenting audio model reasoning.

## 1. Introduction

In recent years, Large Audio Language Models (LALMs) have made substantial progress in foundational perception

[1] Institute of Acoustics, Chinese Academy of Sciences [2]University of Chinese Academy of Sciences [3]University of California, Merced [4] Institute of Computing Technology, Chinese Academy of Sciences [5]The University of Queensland. Correspondence to: Siqian Tong <tongsiqian@mail.ioa.ac.cn>, Xuan Li <lixuan@mail.ioa.ac.cn>.

*Proceedings of the 43rd International Conference on Machine Learning*, Seoul, South Korea. PMLR 306, 2026. Copyright 2026 by the author(s).

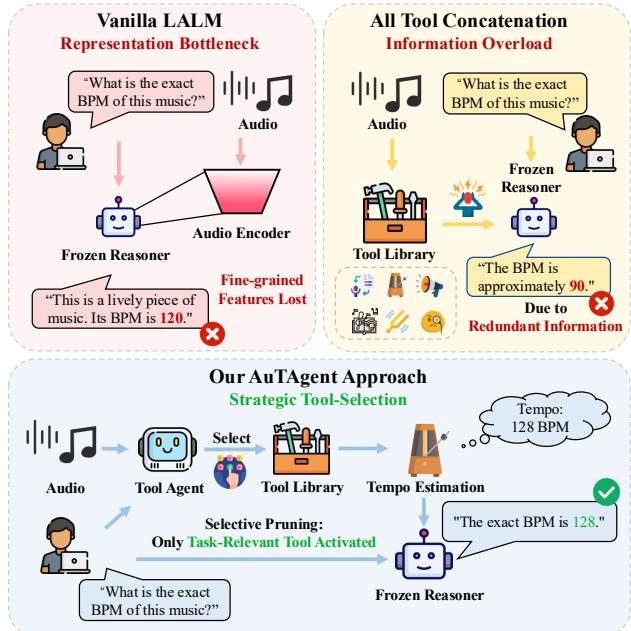

*Figure 1.* Challenges in LALMs and our solution. AuTAgent employs a dynamic tool-selection policy to extract precise evidence (e.g., exact BPM), ensuring accurate and efficient audio reasoning.

tasks such as speech recognition and audio captioning. By aligning audio encoders with Large Language Models (LLMs), LALMs (Tang et al., 2023; Chu et al., 2024; Ghosh et al., 2024; Goel et al., 2025) have demonstrated strong multimodal capabilities. However, while these models can perceive general content, they struggle with complex audio reasoning tasks (Sakshi et al., 2024; Kumar et al., 2025; Yao et al., 2022; Ma et al., 2025) requiring multi-step logical deduction and precise acoustic measurement, for example, analyzing music theory structures or reasoning about the causality of environmental sound events.

This deficiency arises from the representation bottleneck in current architectures, as shown in Figure 1. Most LALMs rely on general-purpose acoustic encoders like Whisper (Radford et al., 2023) (optimized for speech transcription), HuBERT (Hsu et al., 2021) (focused on phonetics), or CLAP (Elizalde et al., 2023) (aligned with global text). These encoders prioritize global semantic information

but often discard fine-grained low-level features (e.g., exact pitch, frequency) and mid-level structures (e.g., chord progressions) during compression. Consequently, due to the scarcity of high-quality fine-grained data, models frequently produce generic descriptions or hallucinate based on textual priors rather than auditory facts (Yuan et al., 2024; Huang et al., 2024b). To break this bottleneck, external tools are essential (Huang et al., 2024a). Tools act as a bridge connecting perception and reasoning, translating opaque audio signals into structured, human-readable features, such as converting vague melodies into specific MIDI notes or transforming background noise into timestamped events. This structured evidence provides a reliable anchor for the model's reasoning process.

Although our preliminary experiments indicate a high theoretical upper bound for tool integration (Theoretical Upper Bound), realizing this potential is non-trivial. Existing strategies, such as zero-shot selection (Yao et al., 2022; Shen et al., 2023) or naive all-tool invocation, often prove counterproductive, performing worse than the no-tool baseline. We attribute this to two factors: blind invocation causes information overload (Shi et al., 2023), where irrelevant tool outputs distract the reasoner; while untrained prompt-based selection introduces misleading information (Yoran et al., 2023), resulting in negative transfer. Audio reasoning follows a dynamic utility distribution: the optimal combination of tools is not static but shifts dynamically with audio complexity and user intent. Relying on simple prompts is insufficient for the model to learn such complex dynamic trade-offs. Only through specialized training to internalize a policy of "when to use which tool" can we achieve precise noise filtering and reasoning enhancement.

To address these challenges, we propose **AuTAgent** (**Au**dio **T**ool **Agent**), a Reinforcement Learning (RL) framework designed for audio tool agents. The core advantage of our approach over existing work lies in solving the dual problem of scarcity of supervised data and uncertainty in tool gain. On one hand, in open-ended reasoning, it is difficult to define a ground-truth tool chain, making traditional supervised learning hard to implement. On the other hand, it is often unclear whether a tool actively contributes to the solution or merely adds noise. AuTAgent adopts a reasoning-performance-driven reinforcement learning paradigm, aiming to achieve autonomous tool-call learning via Group Relative Policy Optimization (GRPO) (Shao et al., 2024). Instead of relying on manual annotation, we utilize feedback from the backend Audio Reasoner to guide the frontend Audio Tool Agent. We introduce a Baseline-Subtracted Differential Reward mechanism: the Agent receives a positive reward only when the tool-assisted result improves upon the base model's direct reasoning. This constraint forces the Agent to suppress redundant operations and focus on strategically mining tools that yield a verifiable net positive gain.

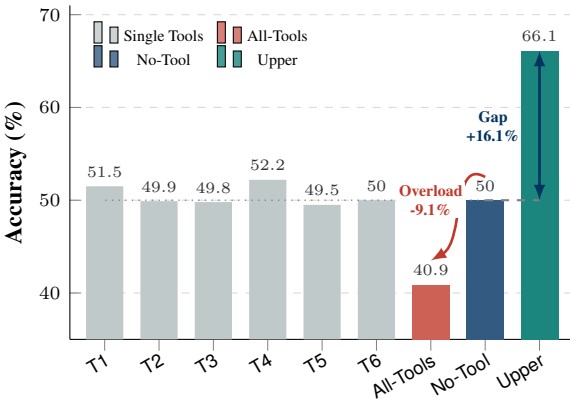

*Figure 2.* Comparison of MMAU Test-mini accuracy across single tools (T1–T6), the All-Tools baseline, the No-Tool baseline and a theoretical upper bound (Upper).

AuTAgent demonstrates high data efficiency and generalization. With approximately 2,000 training samples, it acquires a robust acoustic reasoning paradigm. Experimental results show that AuTAgent improves accuracy on MMAU Test-mini (Sakshi et al., 2024) and MMAR (Ma et al., 2025) by 4.2% and 6.2% for Qwen2-Audio-7B-Instruct; for GPT-4o Audio, the improvements reach 9.8% and 8.0%. Moreover, the learned policy exhibits strong transferability: acting as a plug-and-play module, it empowers closed-source models like GPT-4o Audio to achieve 66.80% on MMAU Test-mini and 62.80% on MMAR. Our proposed framework marks a pivotal paradigm shift from static perception to dynamic, intent-aware active reasoning, establishing a new benchmark for building trustworthy and autonomous multimodal agents. Our main contributions are summarized as follows:

- We identify the representation bottleneck in current LALM architectures that discards fine-grained acoustic details. We propose the AuTAgent framework, which transforms passive tool invocation into an active, context-aware tool-calling process through GRPO training.
- We design a baseline-subtracted differential reward mechanism that effectively addresses redundancy and noise interference in tool selection, achieving simultaneous improvements in both accuracy and reasoning efficiency.
- Extensive experiments demonstrate that AuTAgent not only achieves outstanding performance for open-source models on the MMAU Test-mini and MMAR benchmarks with high data efficiency (∼2k samples), but also significantly enhances the reasoning performance of GPT-4o Audio as a general-purpose plugin.

## 2. Preliminary Analysis: Potential and Pitfalls of Tool Augmentation

To validate the potential of incorporating external tools for audio reasoning and to assess the necessity of an effective tool-selection strategy, we conducted exploratory experiments on the MMAU Test-mini split. We utilized a library

of six distinct audio processing tools (denoted as T1 through T6; detailed specifications are provided in Section 4.4). We established four distinct experimental setups to investigate different paradigms of tool utilization:

**No-Tool Baseline.** The frozen base model, Qwen2-Audio-7B-Instruct, processes audio-query pairs directly without accessing external tools.

**Single-Tool Isolation.** Each of the six tools (T1–T6) is mandatorily applied individually during inference to evaluate their standalone impact.

**All-Tools Baseline.** All six tools are executed simultaneously, and their aggregated outputs are concatenated and fed into the reasoning model.

**Theoretical Upper Bound.** A post-hoc metric counting a sample as correct if any individual tool succeeds, representing the performance ceiling for single-tool selection.

As illustrated in Figure 2, the results reveal two critical phenomena:

**(1) Significant Potential vs. Performance Gap.** A substantial gap exists between the No-Tool Baseline (50.0%) and the Oracle Upper Bound (66.1%). This indicates that the tool library contains complementary knowledge capable of solving complex cases where the base model fails. Bridging this gap requires precise tool invocation.

**(2) Blind Tool Adoption Leads to Degradation.** Simply stacking all tools (All-Tools) results in a severe performance drop to 40.9%, significantly lower than the baseline. This confirms that irrelevant tool outputs introduce noise interference and information overload, distracting the model rather than aiding it. Furthermore, individual tools (T1–T6) exhibit varying performance: while specific single tools (e.g., T4 at 52.2%, T1 at 51.5%) offer marginal gains over the baseline, others (e.g., T2, T3, T5) degrade performance. Crucially, even the best-performing single tool trails the Oracle (66.1%) by a massive margin (∼14%). This demonstrates that no single tool is universally applicable; the optimal tool varies dynamically with the audio-query pair.

These findings demonstrate that while external tools hold immense potential, indiscriminate integration is detrimental. The key to bridging the reasoning gap requires a dynamic tool-selection strategy based on the specific audio-query context, motivating the design of our AuTAgent.

# 3. Method

In this section, we propose AuTAgent, a reinforcement learning framework designed for tool-augmented audio reasoning. Departing from prior approaches that rely on training-free prompting or supervised fine-tuning, AuTAgent enables the audio tool agent to master tool selection through trial-

---

**Algorithm 1** AuTAgent Training Steps with GRPO

---

**Require:** Training Batch $\{(x_{a,i}, x_{q,i}, y_i^*)\}_{i=1}^B$
**Require:** Agent $\pi_\theta(t|s)$, Frozen Reasoner $\mathcal{R}_\phi$, Group Size $G$

1: **for** $i = 1 \ldots B$ **do**
2:     Calculate baseline: $y_{\text{base}} \leftarrow \mathcal{R}_\phi(x_{a,i}, x_{q,i})$
3:     **Group Sampling:**
4:     **for** $g = 1 \ldots G$ **do**
5:         Sample tool: $z^{(g)} \sim \pi_\theta(\cdot \mid x_{a,i}, x_{q,i})$
6:         Execute tool: $o^{(g)} \leftarrow t_{z^{(g)}}(x_{a,i})$
7:         Reason: $\hat{y}^{(g)} \leftarrow \mathcal{R}_\phi(x_{a,i}, x_{q,i}, o^{(g)})$
8:         $r^{(g)} \leftarrow \mathbb{I}(\hat{y}^{(g)} = y_i^*) - \mathbb{I}(y_{\text{base}} = y_i^*)$
9:     **end for**
10:     **Optimization:**
11:     Compute Advantage: $A_i^{(g)} \leftarrow r_i^{(g)}$
12:     Update policy using GRPO objective with $A_i^{(g)}$
13: **end for**

---

and-error interaction. As shown in Figure 3, by leveraging the exploration-exploitation dynamics of reinforcement learning, the agent adaptively selects tools from the tool library based on performance feedback. Notably, the underlying reasoning backbone remains frozen throughout the process. This decoupling allows the learned tool selection policy to be transferable across different reasoning backbones without requiring additional training.

## 3.1. Problem Definition

Formally, we consider instances sampled from a task distribution $\mathcal{D}$, where each instance consists of an input state $s = (x_a, x_q)$, comprising an audio clip $x_a$ and a textual question $x_q$, and a corresponding ground-truth answer $y^*$. We establish a library of $K$ external tools, denoted as $\mathcal{T} = \{t_1, \ldots, t_K\}$, where each tool functions as a specialized module capable of extracting structured knowledge from the audio $x_a$ (see Table 1). The system comprises two decoupled components:

- **Audio Tool Agent (Agent, $\pi_\theta(t|s)$):** A trainable policy model parameterized by $\theta$.
- **Reasoner ($\mathcal{R}_\phi$):** A frozen Large Language Model.

The objective of our AuTAgent framework is to enable the agent to autonomously explore and discover effective tool combinations without relying on explicit ground-truth labels for tool selection. Given a state $s = (x_a, x_q)$, the agent generates a reasoning trajectory followed by a discrete combination of selected tool indices $z \subseteq \{1, \ldots, K\}$. The reasoner then processes the raw audio $x_a$, the query $x_q$, and the tool-augmented output $o_z = t_z(x_a)$ to predict the final answer $\hat{y} = \mathcal{R}_\phi(x_a, x_q, o_z)$.

We optimize the policy by comparing $\hat{y}$ against a baseline prediction $y_{base} = \mathcal{R}_\phi(x_a, x_q)$ derived solely from the raw inputs (optimization details in Section 3.2). The entire train-

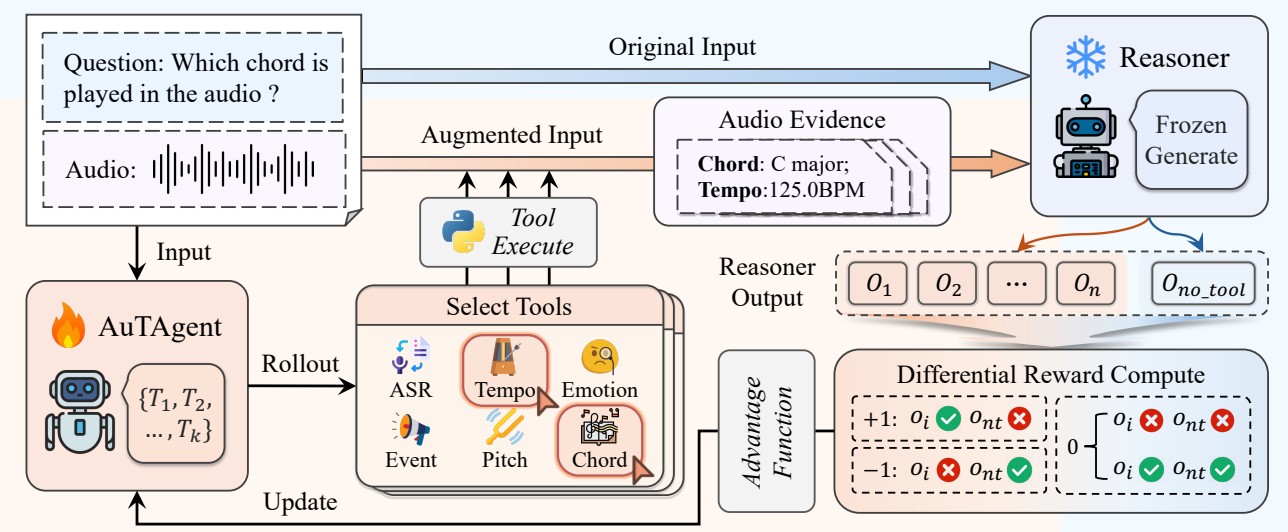

*Figure 3.* Training framework of AuTAgent. Given an audio-query pair, the Audio Tool Agent select specific tools and generate structured audio evidence, which augments the input for a frozen backend Reasoner. By introducing a Baseline-Subtracted Differential Reward mechanism, the agent receives a positive reward only when it successfully corrects baseline errors. This constraint drives the discovery of effective, context-aware tool combinations through GRPO policy updates, effectively suppressing redundancy.

ing process is driven by Group Relative Policy Optimization (GRPO), as outlined in Algorithm 1.

## 3.2. Policy Optimization via Differential Reward Mechanism

To enable the audio tool agent to autonomously discover synergistic tool combinations without explicit supervision, we employ a training framework based on GRPO. Unlike conventional approaches that solely focus on the correctness of the final answer, we propose a Baseline-Subtracted Differential Reward function. The core motivation behind this design is that an optimal tool selection strategy should not merely aim to "answer correctly", but rather to "answer correctly because of the tool usage", thereby quantifying the net gain provided by the tools.

**Differential Reward Formulation.** Formally, given an input state $s = (x_a, x_q)$ consisting of audio $x_a$ and a query $x_q$, we define the reward $r$ as the *net performance gain* attributed strictly to tool usage. Specifically, for each training instance, we first generate a baseline prediction $y_{\text{base}}$ using the frozen reasoner with raw inputs. We then sample $G$ tool selection candidates $\{z_i\}_{i=1}^G$ from the current policy $\pi_\theta$.

To rigorously quantify the utility of the selected tools, we adopt a strict baseline-subtraction mechanism. For each candidate $z_i$, the tool-aware reward $r_i$ is computed as:

$$r_i = \mathbb{I}(\hat{y}_i = y^*) - \mathbb{I}(y_{\text{base}} = y^*) \qquad (1)$$

where $\hat{y}_i$ denotes the prediction generated by the reasoner using the selected tools $z_i$, $y^*$ represents the ground truth,

and $\mathbb{I}(\cdot)$ is the indicator function. By fixing the penalty weight for the baseline to 1, this formulation results in a discrete ternary reward signal $r_i \in \{-1, 0, 1\}$, which guides the policy optimization with clear objectives:

- **Reward = 1 (Critical Assistance):** This maximal reward is granted *only* when the baseline fails ($y_{\text{base}} \neq y^*$) but the tool-augmented path succeeds ($\hat{y}_i = y^*$). This incentivizes the agent to identify and select tools that are strictly necessary for solving complex queries that exceed the baseline's capabilities.
- **Reward = 0 (Suppression of Redundancy):** A zero reward is assigned if the baseline itself is already correct ($y_{\text{base}} = y^*$), even if the tool-augmented prediction is also correct. Unlike traditional RL which might reward any correct answer, our framework treats tool usage on simple queries as a "zero-gain" action. This effectively suppresses the selection of tools for easy samples, encouraging the agent to conserve computational resources and avoid unnecessary complexity.
- **Reward = -1 (Penalty for Distraction):** A negative penalty occurs when the baseline is correct ($y_{\text{base}} = y^*$) but the tool-augmented path leads to an incorrect answer ($\hat{y}_i \neq y^*$). This explicitly signals to the agent that the selected tools introduced noise or misleading information that degraded the reasoner's inherent performance.

**Group Relative Update.** Based on the aforementioned reward, we employ the GRPO algorithm to iteratively update the policy. For the sampled group of $G$ candidates, we first compute the mean and standard deviation of the intra-group rewards. Subsequently, we calculate the group

relative advantage for each candidate:

$$\hat{A}_i = \frac{r_i - \text{mean}\left(\{r_j\}_{j=1}^G\right)}{\text{std}\left(\{r_j\}_{j=1}^G\right)}, \qquad (2)$$

This normalization compares the performance of each tool selection against other candidates, enabling the agent to distinguish which tool combinations perform better for a specific query. Finally, we update the policy parameters $\theta$ by maximizing the following objective function:

$$\mathcal{J}(\theta) = \mathbb{E}_{s \sim \mathcal{D},\, \{o_i\}_{i=1}^G \sim \pi_{\theta_{\text{old}}}(\cdot|s)} \frac{1}{G} \sum_{i=1}^{G} \frac{1}{|o_i|} \sum_{t=1}^{|o_i|} \left\{ \min \right.$$

$$\left. \left[ r_t^{(i)}(\theta) \hat{A}_i,\, \text{CLIP}\left( r_t^{(i)}(\theta), 1 \pm \epsilon \right) \hat{A}_i \right] - \beta D_{\text{KL}}(\pi_\theta | \pi_{\text{ref}}) \right\},$$
$$(3)$$

where $r_t^{(i)}(\theta) = \frac{\pi_\theta(o_i|q, o_{<t}^{(i)})}{\pi_{\theta_{\text{old}}}(o_i|q, o_{<t}^{(i)})}$ serves as an importance sampling ratio. $\pi_{\theta_{\text{old}}}$ is the previous policy. This objective incorporates the clipping mechanism from PPO to ensure training stability and introduces a KL divergence term to constrain the policy from deviating from the reference distribution $\pi_{\text{ref}}$, balancing exploration and exploitation. Through this optimization strategy, the agent learns to select tools that consistently enhance reasoning performance, enabling the autonomous discovery of effective tool-selection strategies.

## 4. Experimental Setup

To comprehensively evaluate the effectiveness and generalization capabilities of AuTAgent in audio reasoning tasks, we conducted experiments according to the following setup.

### 4.1. Dataset Construction

Our training data is derived from the large-scale AVQA Dataset (Yang et al., 2022), widely used for audio-visual question answering. Following the data processing protocol of R1-AQA (Li et al., 2025), we extracted audio tracks from the videos and constructed audio-text pairs by replacing references to "video" in the questions with "audio".

A key distinction in our approach is the data scale. Unlike R1-AQA, which utilizes the full training set (approximately 38k samples), we randomly sub-sampled a micro-training set consisting of only 2,025 samples from the AVQA training split. This minimalist setting is strategically designed to demonstrate that our method does not rely on the rote memorization of massive datasets. Instead, it verifies that the agent effectively learns generalized reasoning paradigms and tool-use strategies via the GRPO mechanism, even under low-resource conditions.

| ID | Tool Module | Implementation Source |
|---|---|---|
| T1 | Automatic Speech Recognition | Whisper-large-v3 (Radford et al., 2023) |
| T2 | Emotion Recognition | emotion2vec_plus_large (Ma et al., 2024) |
| T3 | Chord Recognition | Madmom Library (Böck et al., 2016) |
| T4 | Tempo Estimation | Librosa (Beat Track) (McFee et al., 2015) |
| T5 | Pitch Tracking | Librosa (Piptrack) (McFee et al., 2015) |
| T6 | Sound Classification | AST (Gong et al., 2021) |

*Table 1.* The Modular Tool Library. Implementation sources refer to the specific Python Library or pre-trained models used.

### 4.2. Implementation Details

We employ Qwen2-Audio-7B-Instruct as the backbone policy model for our tool agent. The GRPO training process is implemented using the TRL framework and conducted on 8 NVIDIA A100 GPUs, with each device running a batch size of 8. The model is trained for 200 steps with a learning rate of $1 \times 10^{-6}$ and a temperature of 1.0. To sufficiently explore the policy space and encourage diverse tool interactions during training, we sample $G = 6$ generations for each query to construct the group. The model is optimized using the AdamW optimizer (Loshchilov & Hutter, 2017).

### 4.3. Evaluation Benchmarks

We primarily evaluate model performance based on Multiple-Choice Question (MCQ) accuracy across two diverse and challenging benchmarks specifically designed for complex audio understanding: (i) the MMAU Benchmark (Sakshi et al., 2024) (test-mini split), a comprehensive suite covering diverse audio domains (speech, sounds, music) that demands expert-level knowledge and robust reasoning capabilities; and (ii) the MMAR Benchmark (Ma et al., 2025), which emphasizes deep reasoning in complex mixed-audio scenarios, imposing stringent requirements on acoustic perception precision and logical reasoning chains.

### 4.4. Modular Tool Library

To enhance LALM acoustic perception, we curated a six-tool library (Table 1) covering three domains: Speech, Music, and Environmental. By converting complex non-verbal signals into structured, readable features, we enable the agent to ground its reasoning in explicit evidence.

- **Speech Domain (Linguistic & Paralinguistic).** To fully understand speech, we must decouple content from style. T1 (Automatic Speech Recognition) transcribes spoken content into text, establishing the semantic foundation for linguistic reasoning. Complementing this, T2 (Emotion Recognition) captures paralinguistic nuances(e.g., anger), which are typically lost in pure text transcripts.
- **Music Domain (Harmonic & Rhythmic).** Music reasoning requires analyzing intricate frequency structures and temporal patterns. We integrate three tools to dissect these dimensions: T3 (Chord Recognition) extracts chord progressions to support high-level music theory

| Method | MMAU Test-mini | | | | MMAR | | | |
|---|---|---|---|---|---|---|---|---|
| | Sound | Music | Speech | Avg. | Sound | Music | Speech | Avg. |
| *Open-Source Backbone (Qwen2-Audio-7B-Instruct)* | | | | | | | | |
| Vanilla (No-Tool) | 56.46 | 49.40 | 44.14 | 50.00 | 37.58 | 22.33 | 38.78 | 35.10 |
| Random Selection | 60.96 | 47.90 | 38.44 | 49.10 | 47.27 | 30.10 | 40.82 | 38.60 |
| All-Tools (Concat.) | 55.26 | 35.63 | 31.83 | 40.90 | 40.00 | 25.24 | 37.07 | 35.50 |
| Tools w/ Desc. | **63.06** | 50.30 | 40.24 | 51.20 | 40.00 | 27.67 | 40.00 | 37.50 |
| **AuTAgent (Ours)** | 61.56 | **53.59** | **47.45** | **54.20** | **48.48** | **31.55** | **45.92** | **41.30** |
| *Closed-Source Backbone (GPT-4o Audio)* | | | | | | | | |
| Vanilla (No-Tool) | 58.86 | 56.59 | 56.76 | 57.40 | 43.64 | 35.44 | 67.01 | 55.00 |
| Random Selection | 57.96 | 48.50 | 54.35 | 53.60 | 42.42 | 30.58 | 61.22 | 52.20 |
| All-Tools (Concat.) | 54.35 | 46.11 | 54.65 | 51.70 | 38.79 | 29.61 | 64.29 | 49.70 |
| Tools w/ Desc. | 62.16 | 57.19 | 69.37 | 62.90 | 50.91 | 43.20 | 72.79 | 60.00 |
| **AuTAgent (Ours)** | **66.97** | **61.8** | **72.97** | **67.20** | **51.52** | **45.15** | **73.81** | **63.00** |

*Table 2.* Accuracies(%) comparison on MMAU Test-mini and MMAR, where Vanilla scores are based on our own evaluation. The best-performing methods in each category are highlighted in **bold**, and the second-best scores are underlined.

analysis. T4 (Tempo Estimation) calculates the Beats Per Minute (BPM), providing critical quantitative evidence for judging speed and rhythm. T5 (Pitch Tracking) extracts fundamental frequency ($F_0$) contours to analyze melody lines and pitch stability.

- **Environmental Domain (Acoustic Events).** For general acoustic scenes, T6 (Sound Classification) identifies specific background events, enabling the agent distinguish foreground signals from background noise, which is essential for accurate scene understanding.

## 5. Results and Analysis

### 5.1. Main Results

Table 2 evaluates AuTAgent on the MMAU Test-mini and MMAR benchmark datasets across both open source and close source, providing a comprehensive comparison between our dynamic policy and four different baselines. Among them, No-Tool and All-Tools have been explained in Section 2. Random Selection refers to the Agent (Untrained Qwen2-Audio-7B-Instruct) assisting reasoning by randomly calling tools from the tool library. Tools w/ Desc. is a description-based zero-shot strategy where the agent autonomously decides whether and which tools to call based on the textual descriptions of the tools. Detailed descriptions are provided in Appendix A.1.

The results indicate that regardless of whether the frozen reasoning backbone is open-source (Qwen2-Audio-7B-Instruct) or closed-source (GPT-4o Audio), AuTAgent consistently achieves the best performance. Using Qwen2-Audio-7B-Instruct, it reaches an average accuracy of 54.20% on MMAU Test-mini and 41.30% on MMAR. The performance on GPT-4o Audio improved to 66.80% and 63.00%,

respectively. Compared to the vanilla baseline, this confirms that in audio tasks, particularly in the Music and Speech subtasks, the structured evidence provided by external tools (such as pitch and rhythmic structures in music) helps LALMs overcome the representation bottleneck.

However, not all tool integration methods lead to performance gains. Beyond the noise interference and information overload from All-Tools mentioned in Section 2, when the audio tool agent lacks strategic training, the improvements from Random Selection and the Tools w/ Desc. are minimal or even negative. This suggests that even with clear tool descriptions, LALMs struggle to accurately judge when and which tools are absolutely necessary. In contrast, through GRPO training and the differential reward mechanism, AuTAgent achieves a transition from passive calling to strategic planning. It can not only precisely identify effective tools based on different audio contexts but also learn to suppress redundant calls in simple tasks to avoid noise interference. This enables particularly outstanding performance in tasks that are highly dependent on specific acoustic features.

### 5.2. Transferability Analysis

Table 3 evaluates AuTAgent's transferability by comparing its performance when trained and tested on the same backbone (Self) versus its performance when transferred across different frozen reasoners (Plugin).

Experimental results show that both Self and Plugin settings consistently outperform the Vanilla (No-Tool) baseline. Specifically, without additional training on a specific backbone, the Plugin strategy improves the performance of GPT-4o Audio by 9.4% on MMAU Test-mini and 7.8% on MMAR. Meanwhile, the performance gap between the Self

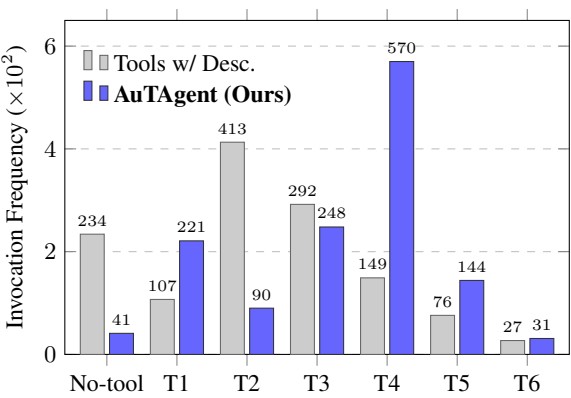

*Figure 4.* Distribution Shift in Tool Invocation frequency across the description-based agent and our RL-trained agent.

| Method | MMAU Test-mini (Avg) | MMAR (Avg) |
|---|---|---|
| **Frozen Reasoner: Qwen2-Audio-7B-Instruct** | | |
| Vanilla (No-Tool) | 50.00 | 35.10 |
| AuTAgent (Self) | **54.20** | **41.30** |
| AuTAgent (Plugin) | 53.60 | 40.50 |
| **Frozen Reasoner: GPT-4o Audio** | | |
| Vanilla (No-Tool) | 57.40 | 55.00 |
| AuTAgent (Self) | **67.20** | **63.00** |
| AuTAgent (Plugin) | 66.80 | 62.80 |

*Table 3.* Comparison of accuracies (%) on MMAU Test-mini and MMAR across open- and closed-source frozen reasoners.

and Plugin is remarkably narrow, remaining within 0.2%–0.8% across all benchmarks and backbones. For example, on GPT-4o Audio, the Plugin policy achieves 66.80% on MMAU Test-mini and 62.80% on MMAR, nearly matching the 67.20% and 63.00% of the AuTAgent(Self) framework.

These data prove that the knowledge of "when to call which tool" learned by AuTAgent is not backbone-specific. The policy captures universal audio reasoning logic rather than over-fitting to the internal representations of a specific reasoner, allowing it to serve as a highly transferable plug-and-play enhancement module. Furthermore, the robust performance of the Plugin strategy demonstrates the feasibility of utilizing open-source, low-cost reasoners (e.g., Qwen2-Audio-7B-Instruct) to train specialized agent and directly deploying them to enhance closed-source models, offering new possibilities for the efficient development and deployment of audio tool agents.

### 5.3. Policy Evolution and Distributional Shift

To quantify the impact of GRPO training, we analyzed the shift distribution in tool usage.

Figure 4 illustrates the shift in tool-call distribution between the description-based zero-shot baseline and the AuTAgent framework. The results indicate that despite having detailed tool descriptions, the untrained model frequently defaults

| Method | Accuracy (%) | | Avg. Tools ↓ |
|---|---|---|---|
| | MMAU | MMAR | |
| Vanilla(No-Tool) | 50.00 | 35.10 | 0.00 |
| Binary Reward | 51.40 | 39.00 | 1.62 |
| **AuTAgent (Ours)** | **54.20** | **41.30** | **1.30** |

*Table 4.* Comparative analysis of Binary and Differential Reward mechanisms. Without the differential penalty, standard RL leads to higher redundancy ("Avg. Tools") and suboptimal performance.

to No-tool (234 calls). AuTAgent reduces this frequency by 82% (down to 41 calls), indicating a learned preference for verifiable acoustic details via external tools. Meanwhile, the policy exhibits significant convergence on T4, with the number of invocations surging from 149 to 570, while the use of suboptimal tools, such as T2 (413 → 90) is significantly reduced. This demonstrates that reinforcement learning enables the agent to transcend textual heuristics and identify tools with actual reasoning gains.

The contrast between the two distributions confirms that static descriptions are insufficient for achieving the optimal selection of tools. AuTAgent effectively aligns tool supply with reasoning demands, allowing the RL-trained agent to prioritize obtaining verifiable evidence over noise, maximizing reasoning precision.

### 5.4. The Necessity of Differential Reward

As summarized in Table 4, we assess the necessity of our reward design via a Binary Reward variant, which provides a constant positive reward ($r = 1$) based solely on the correctness of the tool-augmented output, regardless of the base model's independent performance.

Experimental results indicate that while the Binary Reward variant achieves marginal improvements in accuracy over the vanilla baseline, it is consistently surpassed by AuTAgent in both accuracy and efficiency. This variant exhibits a higher average tool invocation frequency (1.62 vs. 1.30). This confirms that standard reinforcement learning tends to induce "shortcut learning", where the agent performs redundant tool calls even for simple queries already solvable by the base model to secure rewards.

In contrast, our differential reward mechanism explicitly penalizes zero-gain operations by assigning zero reward when the baseline prediction is already correct. This optimization objective drives the policy toward a more parsimonious approach. This mechanism effectively filters out redundant informational noise, allowing the model to reach a higher reasoning performance ceiling through strategic tool use.

### 5.5. Case Study
Here we show relevant cases for different domains of audio. As shown in Table 5, these cases demonstrate the audio

| Qwen2-Audio-7B-Instruct(Vanilla) | AuTAgent (Ours) | |
|---|---|---|
| Wrong Answer | Tool Call | Final Answer |
| **Question 1 (Music):** What is the root chord that starts at 10.14 seconds in the audio? [Ground Truth: A#:maj] | | |
| The audio has a bright, acoustic pop feel. The chord at 10 seconds is likely G major. | `Chord_Recognition:` {[10.14s - 12.50s]: **A#:maj**} | The root chord starting at 10.14s is an A#:maj. |
| **Question 2 (Speech):** How many times does the word "ninety-nine" appear in the sentence? [Ground Truth: three] | | |
| The speaker mentions the number "ninety-nine" twice. | `Auto_Speech_Recognition:` {99 and 99 added is twice times 99.} | The speaker explicitly repeats "ninety-nine" three times throughout the sentence. |
| **Question 3 (Sound):** What could have caused the man's speech to be interrupted? [Ground Truth: the ringing of church bells] | | |
| The man stops speaking due to a sudden loud clap, possibly from an audience. | `Sound_Classification:` {Speech, **Church bell**} | The man's speech was interrupted by the ringing of church bells loudly in the background. |

*Table 5.* Case study comparing vanilla Qwen2-Audio-7B-Instruct and AuTAgent. The table demonstrates how AuTAgent overcomes the "representation bottleneck" by dynamically selecting specialized tools to provide structured acoustic evidence for reasoning.

reasoning capabilities of AuTAgent. AuTAgent can strategically call tools for different audio-question pairs to compensate for the deficiencies in the reasoning of LALMs: **Music:** By calling the Chord_Recognition tool, AuTAgent obtains precise structured frequency features and thus gave the correct answer. **Speech**: AuTAgent utilizes ASR tool to transform speech signals into structured text, correcting the count from "two" to "three". **Sound:** The Sound_Classification tool is called to parse overlapping or low-level acoustic details. AuTAgent provides the missing domain-specific knowledge required to reach the correct conclusion.

## 6. Related Work

**Large Audio Language Models** LALMs have advanced significantly by integrating audio encoders with Large Language Models (LLMs). Representative works such as SALMONN (Tang et al., 2023) and Qwen-Audio (Chu et al., 2023) leverage large-scale audio-text pairs to achieve strong performance in perception tasks like speech recognition and audio captioning. Notably, Qwen2-Audio (Chu et al., 2024) further enhances interaction capabilities through instruction tuning. However, most existing LALMs face a representation bottleneck. Standard acoustic encoders (Radford et al., 2023; Hsu et al., 2021; Elizalde et al., 2023) tend to prioritize global semantic information at the expense of fine-grained acoustic details, such as pitch, or chord structures (Yuan et al., 2024). This loss of information often leads to hallucinations or errors when models attempt tasks requiring precise acoustic measurement or complex reasoning. These limitations highlight the need for external tools to compensate for encoder deficiencies.

**Tool-Augmented Multimodal Agents** Tool learning addresses the limitations of internal parametric knowledge. In textual and visual domains, methods like Toolformer (Schick et al., 2023) and ChartAgent (Wang et al., 2025) have demonstrated the effectiveness of learning to use APIs or decomposing tasks into executable steps. Simi-

larly, VisTA (Huang et al., 2025) explores optimizing tool usage strategies via reinforcement learning rather than static prompting. In contrast, tool-augmented agents in the audio domain remain largely heuristic. Existing frameworks, such as AudioGPT (Huang et al., 2024a) and HuggingGPT (Shen et al., 2023), typically rely on training-free paradigms (e.g., ReAct (Yao et al., 2022)) that use in-context learning to schedule off-the-shelf models. However, indiscriminate tool invocation can introduce irrelevant noise (information overload), while untrained prompt-based selection may lead to negative transfer (Yoran et al., 2023). Moreover, the resulting redundant outputs may undermine the reasoner's faithful grounding on relevant context (Bi et al., 2024). Without a mechanism to learn dynamic trade-offs for tool selection, these methods struggle in complex acoustic scenarios.

**Reinforcement Learning for Reasoning & Tool Use** Reinforcement Learning (RL) has proven effective for enhancing reasoning in LLMs. Recent work such as DeepSeek-R1 (Guo et al., 2025) demonstrates that RL can improve logical reasoning by encouraging the generation of extensive Chains of Thought (CoT). RGR-GRPO (Bi et al., 2025) further advances GRPO by introducing rubric-driven dense rewards for multi-domain reasoning. In the audio domain, R1-AQA (Li et al., 2025) applies GRPO to LALMs, showing that RL outperforms Supervised Fine-Tuning (SFT) in audio question answering. Additionally, Audio-Thinker (Wu et al., 2025) uses RL to guide the reasoning process. Unlike these approaches, which focus on optimizing the internal reasoning chain, AuTAgent applies RL to the external action space. AuTAgent leverages GRPO to autonomously discover optimal tool composition paths, addressing the dual challenges of uncertain tool gain and the scarcity of supervised data.

## 7. Conclusion

In this work, we present AuTAgent, a reinforcement learning framework designed for autonomous, tool-augmented audio reasoning. To mitigate the inherent representation bot-

tlenecks in conventional LALMs and the noise interference stemming from improper tool integration, AuTAgent introduces a baseline-subtracted differential reward mechanism within a GRPO framework. This design enables the agent to discover effective and context-aware tool combinations through trial-and-error interaction. Empirical evaluations demonstrate that AuTAgent significantly enhances accuracy, achieving gains of 4.2% and 6.2% on MMAU Test-mini and MMAR for open-source backbones, and more substantial improvements of 9.8% and 8.0% on closed-source models such as GPT-4o Audio. Crucially, the decoupled design of AuTAgent ensures that the learned policy is transferable, serving as a plug-and-play module that empowers diverse reasoning backbones without requiring additional training. This work establishes a robust, model-agnostic paradigm for reliable and efficient complex audio reasoning.

## Acknowledgements

This work is supported in part by the National Natural Science Foundation of China under Grant Nos. 62471463.

## Impact Statement

This paper presents work whose goal is to advance the field of Machine Learning. There are many potential societal consequences of our work, none which we feel must be specifically highlighted here.

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

# A. Prompt Details

Our prompt engineering follows a systematic methodology characterized by three core pillars: role-specific task definition, modular information integration, and stringent structural constraints. For instance, during the tool selection phase, we explicitly define the agent's domain expertise and provide a comprehensive functional library for audio processing, encompassing dimensions such as speech transcription and tempo estimation. Finally, we enforce precise output formats through the use of dedicated XML-style tags (e.g., <answer></answer>). By maintaining this rigorous, modular structure, we have established a scalable framework that can be effectively generalized across both open-source and proprietary large audio-language models (ALMs). The following sections detail the primary prompts utilized throughout the various stages of the AuTAgent framework:

## A.1. Tools w/Desc. Prompt

You are an expert agent specialized in selecting tools to solve **audio reasoning tasks**. You are provided with access to **6 tools**, indexed from **0 to 5**. Each tool is implemented differently. Treat all tools as **independent**.

**Function:**
 0: Automatic Speech Recognition Tool
   - Description: Transcribes spoken content into text. It is capable of performing multilingual speech recognition, speech translation, and language identification.
 1: Emotion Recognition Tool
   - Description: It is strictly designed to classify the speaker's emotional state (e.g., happy, sad, angry, fearful, neutral) and sentiment from audio signals.
 2: Chord Recognition Tool
   - Description: Predict the harmonic structure and chord progression (e.g., C major, A minor, G7) of a musical piece by analyzing chroma vectors extracted from the audio signal.
 3: Beat and Tempo Tracking
   - Description: It estimates the global tempo (BPM) and detects the beat events (time locations of beats) to analyze the rhythmic structure of the audio.
 4: Pitch Tracking
   - Description: Pitch tracking on thresholded parabolically-interpolated STFT. This tool estimates the fundamental frequency (f0) of the audio over time, useful for analyzing melody, intonation, or tonal characteristics.
 5: Audio Event Detection Tool
   - Description: It is designed to classify and tag audio events, ranging from environmental sounds to human non-speech sounds.

**Query: question here**

Your job:
  1. Carefully analyze the **Audio content** and the **Query**. Select the **index number(s)** of the tools that are **most helpful** for solving the task.
  2. You MUST output **only the selected tool indices** as a comma-separated list, enclosed in <answer></answer> tags.

## A.2. AuTAgent Prompt

You are an expert agent specialized in selecting tools to solve **audio reasoning tasks**. You are provided with access to **6 tools**, indexed from **0 to 5**. Each tool is implemented differently. Treat all tools as **independent**.

**Function:**
   0: type1 (A)
   1: type2 (B)
   2: type3 (C)
   3: type4 (D)
   4: type5 (E)
   5: type6 (F)

**Query: question here**

Your job
   1. Carefully analyze the **Audio content** and the **Query**. Select the **index number(s)** of the tools that are **most helpful** for solving the task.
   2. You MUST output **only the selected tool indices** as a comma-separated list, enclosed in <answer></answer>tags.

## A.3. Tool-Augmented Reasoner Prompt

Question:  **question here**

Extra information from the audio analysis:  **chart-table-info here**

Answer the question based on the audio and extra information. You MUST output your final answer ONLY within <answer></answer>tags. Be concise. Example:<answer>dog</answer>.

## A.4. Baseline Reasoner Prompt

Question: question here

Answer the question based on the audio and extra information. You MUST output your final answer ONLY within <answer></answer>tags.

# B. Tool Output Json Format

## B.1. Automatic Speech Recognition Tool Output Format

```
{
    "tool":  "automatic-speech-recognition",
    "output":  "The station is a bit of a mess, but it's not too bad."
}
```

**B.2. Emotion Recognition Tool Output Format**

```
{
    "tool":  "emotion recognition",
    "output":  "sad"
}
```

**B.3. Chord Recognition Output Format**

```
{
    "tool":  "chord_recognition",
    "output":  [
      {
        "timestamp":  [0.0, 2.9],
        "value":  "A:maj"
      },
      {
        "timestamp":  [2.9, 6.6],
        "value":  "G:maj"
      },
      {
        "timestamp":  [6.6, 9.5],
        "value":  "D:maj"
      }
    ]
}
```

**B.4. Tempo Estimation Tool Output Format**

```
{
    "tool":  "Tempo Estimation",
    "output":  "117.19"
}
```

## B.5. Pitch Tracking Tool Output Format

```
{
    "tool":  "Pitch Tracking",
    "output":  [
      {
        "timestamp":  0.0,
        "fundamental frequency":  "520.91 Hz"
      },
      {
        "timestamp":  1.504,
        "fundamental frequency":  "328.85 Hz"
      },
      {
        "timestamp":  3.008,
        "fundamental frequency":  "420.01 Hz"
      },
      {
        "timestamp":  4.512,
        "fundamental frequency":  "400.14 Hz"
      },
      {
        "timestamp":  6.016,
        "fundamental frequency":  "701.12 Hz"
      },
      {
        "timestamp":  7.52,
        "fundamental frequency":  "1558.71 Hz"
      },
      {
        "timestamp":  9.024,
        "fundamental frequency":  "1025.21 Hz"
      }
    ]
}
```

## B.6. Sound Classification Tool Output Format

```
{
    "tool":  "Sound Classification",
    "output":  "Basketball bounce, Walk, footsteps, Speech"
}
```

