# OpenReview forum: "AuTAgent: A Reinforcement Learning Framework for Tool-Augmented Audio Reasoning"
_ICML.cc/2026/Conference — ICML 2026 regular_

### Official Review · Reviewer_1neR · 2026-02-28

**Soundness:** 3
**Presentation:** 3
**Significance:** 2
**Originality:** 2
**Overall Recommendation:** 4
**Confidence:** 4

**Summary:**

This paper investigates reinforcement learning-based tool selection for large audio language models (LALMs) and proposes AuTAgent, a framework that trains a tool-selection agent to decide when and which external audio processing tools should be invoked to improve audio understanding and reasoning performance. The method is motivated by the preliminary results that using all tools hurts the performance while selecting the optimal tool can substantially enhances the performance.
The key technical contribution is a baseline-subtracted differential reward mechanism within GRPO, where tool usage is rewarded only if it improves performance over a frozen backbone’s baseline prediction, to penalize unnecessary tool invocation. Experiments show consistent gains over baselines, and the learned policy transfers effectively across open-source and closed-source backbones.

**Compliance With Llm Reviewing Policy:**

Affirmed.

**Final Justification:**

See the rebuttal acknowledgement.

**Key Questions For Authors:**

1. How will altering the tool libraries affect the results? Intuitively LALMs excel at high-level audio understanding but are not sensitive to low-level details so will using only low-level audio processing tools achieve similar results?
2. How does the tool selecting agent perform on advanced MLLMs, like Gemini-Pro?

**Limitations:**

The paper does not include discussion of limitations. One limitation may be the sensitiveness to the tool library.

**Strengths And Weaknesses:**

Strengths

1. The paper clearly identifies the challenge, the preliminary results and the motivation for the reward design.
2. Effective improvement across backbones: The method is evaluated on both open-source and closed-source models, and the plugin transfer results are particularly compelling. The small performance gap between “Self” and “Plugin” settings strengthens the claim of backbone-agnostic policy learning. The training data with only ~2k samples achieves measurable improvements.
3. Comprehensive analysis. Ablations (binary reward vs. differential reward), distributional analysis of tool invocation, and qualitative case studies help interpret the learned policy.

Weaknesses

1. Incremental novelty. While the application to audio is interesting, the overall framework is conceptually close to existing tool-augmented agents in other fields. The main novelty lies in adapting it to audio and introducing a baseline-subtracted reward, which may be viewed as incremental.
2. Limited backbones. Only one open-source and close-source models are incorporated. The performance of transferring to more models are lacking.
3. Tool library is fixed and small. The experiments rely on a curated set of six tools. It remains unclear how the framework scales to larger or smaller tool libraries. As Figure 4 illustrates, different tools are invoked with varying frequencies. It is unclear whether retaining only frequently-called tools will influence results.

---

> ### Author Rebuttal · Authors · 2026-03-30
>
> **Response to W1: Incremental Novelty & Modality Gap**
>
> We sincerely thank the reviewer. While our RL framework draws inspiration from visual and text tool agents, the audio domain is fundamentally more complex, making our framework far from a mere incremental adaptation.
>
> 1. **The Unique Complexity of Audio:** Unlike discrete text or spatially explicit vision, audio is a continuous, opaque time series. When LALMs process this signal, waveforms are compressed into tokens via mel-spectrograms and acoustic encoders. This creates a severe representation bottleneck: global semantics survive, but fine-grained acoustic details (precise pitch, temporal boundaries) are heavily compressed. Consequently, LALMs inevitably hallucinate when complex reasoning requires precise acoustic measurements.
> 2. **Bridging the Gap:** Because LALMs cannot natively perceive these lost details, our framework bridges the critical gap between coarse semantic perception and fine-grained acoustic reasoning. While visual tools extract features from explicit 2D planes, audio tools face the harder challenge of extracting structured time-frequency features from opaque signals to provide reliable evidence. *(For a broader comparison with visual agents, please see our response to Reviewer Udqf, Point 1).*
>
> **Response to W2: Limited Backbones & Transferability**
>
> We completely agree that validating across broader models is crucial. To address this, we conducted comprehensive supplementary experiments using the prominent closed-source model, **Gemini 2.5 Pro**, evaluating our framework under two distinct configurations to test flexibility:
>
> * **AuTAgent (Plugin):** We directly applied the tool-selection policy (trained on Qwen2-Audio) to Gemini 2.5 Pro without extra training. This tests zero-shot transferability across entirely different LALM architectures.
> * **AuTAgent (Self):** We retrained the policy from scratch, explicitly using Gemini 2.5 Pro as the backend reasoner to provide reward feedback during GRPO. This evaluates the framework's adaptability for optimization on a specific closed-source LLM.
>
> | Method | MMAU Test-mini (%) | MMAR (%) |
> | :--- | :--- | :--- |
> | Gemini 2.5 Pro Vanilla | 66.10 | 65.90 |
> | Gemini 2.5 Pro(Plugin) | 76.50 | 74.10 |
> | Gemini 2.5 Pro(Self) | **77.20** | **75.00** |
>
> Our paradigm consistently and significantly boosts Gemini 2.5 Pro. The strong **Plugin** performance proves AuTAgent captures universal acoustic reasoning regularities rather than overfitting to one backbone. The **Self** setting further maximizes performance, confirming our GRPO framework is entirely **model-agnostic**—seamlessly utilizing different reasoners to construct differential rewards. *(Detailed metrics are available in the link: https://anonymous.4open.science/r/AuTAgent1/Reviewer_1neR_detailed_results.md).*
>
> **Response to W3: Tool Library Scalability & Curation**
>
> 1. **Framework Scaling:** Mathematically, AuTAgent treats the library as a discrete action space $z \subseteq \{1,...,K\}$. Scaling to a larger set (e.g., $K=20$) simply requires adding tools to the environment and updating text descriptions. GRPO autonomously explores this expanded space using differential rewards to discover utility, requiring zero manual trajectories or rule-based tuning.
> 2. **Ablation on Tool Curation:** To answer whether retaining only frequently-called tools suffices, and to test the intuition that "LALMs only need low-level tools since they already possess high-level semantic perception," we conducted targeted ablations. Following Fig 4, high-frequency tools are T4 (Tempo), T3 (Chord), and T1 (ASR). We also split tools into high-level semantics (T1, T2, T6) and low-level acoustics (T3, T4, T5).
>
> | Tool Library Configuration (Qwen2-Audio-7B) | MMAU Test-mini (%) | MMAR (%) |
> | :--- | :--- | :--- |
> | Vanilla Baseline (No-Tool) | 50.00 | 35.10 |
> | Low-level Tools Only (T3, T4, T5) | 51.60 | 38.10 |
> | High-frequency Tools Only (T1, T3, T4) | 52.50 | 39.80 |
> | Full Library (All 6 Tools) | **54.20** | **41.30** |
>
> * **Low-level Only:** Performance drops (51.60%). While LALMs have macro-semantic perception, they struggle with precise symbol extraction in complex environments. High-level tools provide explicit, verifiable anchors that prevent hallucinations in chaotic scenes.
> * **High-frequency Only:** Retaining only frequent tools also degrades performance (52.50%). Audio reasoning is highly diverse. While Tempo (T4) is broadly applicable, lower-frequency tools like Pitch Tracking (T5) are absolutely essential for specialized queries. Removing them artificially caps the agent's upper bound on specialized tasks.
>
> Overall, maintaining a full library spanning low/high-level and frequent/specialized capabilities is indispensable. AuTAgent effectively navigates this diversity, selectively activating tools based on unique contexts. *(Detailed evaluations are provided in the link: https://anonymous.4open.science/r/AuTAgent1/Reviewer_1neR_detailed_results.md).*

---

> > ### Author Rebuttal · Reviewer_1neR · 2026-04-07
> >
> > Thanks for the additional experiments. My concerns have been addressed and I will keep my score.

---

### Official Review · Reviewer_ctLW · 2026-02-28

**Soundness:** 3
**Presentation:** 3
**Significance:** 2
**Originality:** 2
**Overall Recommendation:** 4
**Confidence:** 4

**Summary:**

This paper proposes AuTAgent, a reinforcement learning framework for tool-augmented audio reasoning. The core idea is to train a tool-selection policy using GRPO with a baseline-subtracted differential reward, where the agent is rewarded only when tool usage strictly improves over a frozen reasoning backbone. The method aims to address the representation bottleneck in large audio-language models (LALMs) by selectively invoking specialized audio tools (e.g., ASR, chord recognition, tempo estimation).

Experiments on MMAU Test-mini and MMAR demonstrate consistent gains over no-tool, all-tool, and zero-shot description-based baselines, for both open-source (Qwen2-Audio-7B-Instruct) and closed-source (GPT-4o Audio) backbones (Table 2, p.6). The learned policy also transfers across backbones with minimal degradation.

**Compliance With Llm Reviewing Policy:**

Affirmed.

**Final Justification:**

The author's response has partly addressed my concern, and my score remain positve for acceptance of the paper

**Key Questions For Authors:**

Have you compared AuTAgent against supervised fine-tuning (SFT) or DPO-style preference learning for tool selection under the same 2k training samples? If so, how do the results compare?

How does performance change as the number of tools increases (e.g., by adding redundant or partially overlapping tools)? Does the learned policy remain stable?

Can the framework be extended to multi-turn or sequential tool invocation (e.g., selecting tools conditioned on intermediate tool outputs)? If so, what modifications would be required?

**Limitations:**

There is no limitation section. Consider addressing some of my points on weakness as limitations (e.g., small scale of tool library, limited to single-turn setting, etc.,)

**Strengths And Weaknesses:**

I think the paper is well executed overall. The differential reward design is clean and intuitive — rewarding only net gains over the baseline makes a lot of sense in this setting. The ablation against a binary reward supports that this choice matters. Empirically, the evaluation is fairly thorough. The authors compare against multiple reasonable baselines and show consistent improvements across two benchmarks and both open- and closed-source backbones.

As for the weakness, my main concern is scale. The tool library is relatively small (six tools), and the tools are fairly distinct. It’s unclear whether the approach would still work cleanly with a larger or more overlapping tool ecosystem. The setting is also single-turn MCQ QA. The paper argues for dynamic tool selection, but the experiments don’t really test long-horizon or multi-step tool composition, where RL might shine more clearly.

Conceptually, the method is an adaptation of existing GRPO-style RL to tool selection. The differential reward is sensible, but the novelty is incremental relative to prior RL-for-tool-use work.

---

> ### Author Rebuttal · Authors · 2026-03-30
>
> **Response to Point 1: Comparison with SFT and DPO Paradigms**
>
> We highly appreciate this suggestion. To demonstrate our RL framework's necessity, we evaluated AuTAgent against SFT and DPO under the same 2,025 samples using Qwen2-Audio-7B-Instruct *(Detailed evaluation tables are in the link: https://anonymous.4open.science/r/AuTAgent1/Reviewer_ctLW_detailed_results.md)*.
>
> To ensure fairness, we constructed offline datasets via exhaustive evaluation of all $2^6=64$ tool combinations to reflect our differential reward logic: If the baseline is correct, the optimal policy avoids redundancy; thus, the SFT golden label and DPO $y_{chosen}$ are No-Tool. If the baseline fails, we apply a *Minimal Toolset Criterion*: the combination correcting the error with the absolute minimum tools becomes the SFT golden label and DPO $y_{chosen}$. For DPO's $y_{rejected}$, we sample combinations that either fail or achieve the same result but use strictly more tools.
>
> | Method | Training Data | MMAU Test-mini (%) | MMAR (%) |
> | :--- | :--- | :--- | :--- |
> | Vanilla (No-Tool) | 0 (Zero-shot) | 50.00 | 35.10 |
> | SFT Baseline | 2,025 (Offline Golden Labels) | 52.70 | 39.20 |
> | DPO Baseline | 2,025 (Offline Preference Pairs) | 53.20 | 39.40 |
> | **AuTAgent (Ours)** | **2,025 (Online GRPO)** | **54.20** | **41.30** |
>
> Despite exhaustive data, offline paradigms fall short due to algorithmic bottlenecks:
> * **SFT (Shortcut Learning):** LLMs often bypass explicit evidence, relying on internal priors to guess correctly. SFT blindly clones offline trajectories, forcing models to memorize spurious correlations rather than true causal utility, limiting generalization.
> * **DPO (Exploration Ceiling):** DPO's cognitive boundaries are locked by predefined datasets. Lacking real-time interaction, it cannot explore the $2^6$ action space; if the global optimum is omitted during data construction, DPO never discovers it.
> * **GRPO (Causal Optimization):** AuTAgent dynamically generates tool combinations, executing them for instant feedback via our differential reward. This optimizes directly for verifiable net gains, successfully suppressing redundancies.
>
> **Response to Point 2: Tool Library Scalability and Curation**
>
> To test scalability, we expanded our 6-tool library to a noisier 10-tool library, injecting 2 redundant tools (**Wav2Vec 2.0** overlapping with Whisper; **BeatNet** overlapping with Tempo Estimation) and 2 new tools (**HTDemucs** for source separation; **Pyannote.audio** for speaker diarization). This exponentially expanded the action space from $2^6$ to $2^{10}$.
>
> | Tool Library Configuration | MMAU Test-mini (%) | MMAR (%) | Training Steps |
> | :--- | :--- | :--- | :--- |
> | Vanilla (No-Tool) | 50.00 | 35.10 | 0 |
> | **AuTAgent (Original 6 Tools)** | 54.20 | 41.30 | **200** |
> | AuTAgent (Expanded 10 Tools) | **54.50** | **41.70** | 1,400 |
>
> This test *(detailed data in the link: https://anonymous.4open.science/r/AuTAgent1/Reviewer_ctLW_detailed_results.md)* revealed:
> 1) **Optimal Trade-off:** The 10-tool setup yielded a slight gain (+0.3% on MMAU), but the 16-fold action space increase required 7x more GRPO training steps (from 200 to 1,400) to converge. Our 6-tool library is the optimal engineering balance between reasoning performance and computational cost.
> 2) **Robustness to Redundancy:** Within this massive $2^{10}$ space, the agent still avoided information overload. Driven by the differential reward, the policy inherently learned to filter zero-gain redundant calls and autonomously discover the true causal utility of new tools.
>
> **Response to Point 3: Extension to Multi-turn Sequential Calling**
>
> We fully agree that sequential tool combinations represent a fascinating direction. However, our current concurrent calling scheme strategically addresses the most critical crisis facing LALMs: the Representation Bottleneck. Without reliable acoustic perception, multi-turn logical reasoning will be built upon hallucinations, leading to cascading errors. Therefore, AuTAgent prioritizes establishing reliable structural anchors upfront.
>
> To support sequential calling in future iterations, the RL framework requires the following modifications:
> 1) **Dynamic State Formulation:** Evolving the input state $s_t$ into a dynamic history $s_t = (x_a, x_q, a_{<t}, o_{<t})$.
> 2) **MDP Modeling:** Shifting to a sequential Markov Decision Process (MDP) and introducing a specific termination token to allow the agent to autonomously end the information-gathering phase.
> 3) **Reward Shaping:** Introducing minor negative step penalties to prevent infinite loops and encourage the most streamlined and efficient reasoning paths.

---

> > ### Author Rebuttal · Reviewer_ctLW · 2026-04-04
> >
> > I want to thank the authors for the response, which partially addresses my concern over scale and helps verify the effectiveness against SFT and DPO. However, expanding from 6 to 10 tool settings is still not a large tool library, and there is no actual long-horizon task involved. Thus, I will keep my score (and it is still overall positive towards acceptance).

---

> > > ### Author Response · Authors · 2026-04-04
> > >
> > > **Response to Reviewer ctLW**
> > >
> > > We would like to sincerely thank you for the overall positive assessment (Score 4) and for recognizing that our response effectively addresses the concerns regarding experimental scale and the performance gains over SFT and DPO baselines.
> > >
> > > Regarding the remaining comments on the tool library size and long-horizon tasks:
> > >
> > > * **On the Scale of the Tool Library (6 to 10 Tools):** We appreciate the reviewer's insight. While we have expanded the tool settings from 6 to 10 to provide a more comprehensive evaluation, we acknowledge that this does not yet represent a "massive" tool library. However, the primary focus of **AuTAgent** is to pioneer a specialized **Reinforcement Learning paradigm for the audio domain**. Our core contribution lies in demonstrating how agents can achieve self-evolution through audio-specific feedback loops and perception-action coordination.
> > >
> > > * **On Long-Horizon Tasks:** We fully agree that long-horizon tasks are essential for evaluating system scalability. In this study, we deliberately focused on tasks within a manageable horizon to rigorously isolate and verify the effectiveness of our proposed learning mechanism. Truly long-horizon scenarios introduce additional complexities, such as cumulative error and long-term memory management. We view these as part of the "scaling-up" research phase, whereas the current paper serves as a crucial proof-of-concept for audio-driven agent evolution.
> > >
> > > **Future Work:** We are inspired by these suggestions and intend to explore the integration of larger toolsets and the challenges of long-term planning in our subsequent research to further push the boundaries of AuTAgent.
> > >
> > > Thank you again for the constructive criticism and for supporting the acceptance of our research.

---

### Official Review · Reviewer_Udqf · 2026-03-10

**Soundness:** 2
**Presentation:** 3
**Significance:** 2
**Originality:** 2
**Overall Recommendation:** 3
**Confidence:** 3

**Summary:**

To better enable external audio tools to assist a reasoner in audio analysis, this paper proposes the reinforcement-learning framework AuTAgent. This paper trains the tool agent using the GRPO algorithm together with a differentiable reward mechanism. Experiments on MMAU Test-mini and MMAR demonstrate that AuTAgent, by selecting appropriate tools, effectively improves the audio-reasoning capabilities of both open- and closed-source backbones (Qwen2-audio-7b-instruction and GPT-4o Audio).

**Compliance With Llm Reviewing Policy:**

Affirmed.

**Final Justification:**

The rebuttal has partially addressed my concerns. The authors clarified the differences between audio analysis and visual analysis, and explained the importance of audio tool selection, thereby demonstrating the significance of the research. Regarding the algorithm, the main difference between AuTAgent and tool selection algorithms in the vision domain lies in the reward design. Ultimately, I have raised my score to 3.

**Key Questions For Authors:**

See weaknesses above. If I have misunderstood any aspect of the paper, please correct me. If the authors can address these weaknesses, I would be very willing to increase my score.

**Limitations:**

Yes.

**Strengths And Weaknesses:**

### Strengths:
1. The paper addresses a clear core problem: how to automatically select appropriate audio analysis tools for audio reasoning tasks.
2. The authors design a Differential Reward mechanism to quantify the utility of selected tools.
3. The paper demonstrates efficient training on a 2,025-sample micro training set and visually illustrates the impact of RL on tool selection via tool-usage frequency.

### Weaknesses:
1. The reinforcement-learning training framework and reward design are highly similar to those used in VisualToolAgent (VisTA) [1], which was developed to train visual tool agents. Both approaches use a GRPO-based RL framework and a tool-aware reward.
2. The baselines in the comparative experiments only report the same backbones under different tool configurations, lacks comparisons against other algorithms.
3. I have concerns about the random sampling of training data described in Section 4.1. Modern research trends emphasize scaling up; why choose to intentionally train on a small randomly sampled subset? It would be better to provide experiments using larger amounts of training data.

[1] VisualToolAgent (VisTA): A Reinforcement Learning Framework for Visual Tool Selection. (Arxiv, 2505.20289).

---

> ### Author Rebuttal · Authors · 2026-03-30
>
> **Response to Weakness 1: Similarity to VisualToolAgent (VisTA)**
>
> We sincerely thank the reviewer for the detailed review. While our RL framework is inspired by visual tool-use agents, we respectfully emphasize that the audio and visual domains present fundamentally distinct and complex challenges:
>
> 1. **The Representation Bottleneck:** LALMs compress continuous 1D waveforms into 2D acoustic features (e.g., Mel spectrograms), feeding them into encoders that prioritize global semantic information at the expense of fine-grained acoustic details (pitch, temporal events). This specific bottleneck makes audio reasoning inherently prone to hallucinations.
> 2. **Distinct Action Spaces:** Visual tools extract explicit spatial targets from 2D planes. In contrast, audio tools must extract implicit, non-linguistic physical features from opaque time series, translating them into structured symbols comprehensible to LLMs.
> 3. **Pioneering Feasibility Validation:** AuTAgent is the first RL-based dynamic tool-calling framework for LALMs. It validates whether an RL agent can successfully weigh and utilize time-dependent acoustic tools, laying a novel foundation for shifting LALMs from static perception to verifiable, active reasoning.
>
> **Response to Weakness 2: Lack of Baseline Comparison**
>
> We highly appreciate the suggestion. As AuTAgent is the first RL-based dynamic tool selection strategy for LALMs, direct baselines are absent. However, we agree that comparing against established paradigms is vital.
>
> To address this, we evaluated AuTAgent against **Supervised Fine-Tuning (SFT)** and **Direct Preference Optimization (DPO)**. *(For the specific experimental design of SFT and DPO, please refer to our response to Reviewer ctLW, Point 1).*
>
> As shown below, standard paradigms fall short. SFT tends to blindly mimic trajectories rather than understanding true causal utility, which severely limits its generalization capabilities on unseen data. DPO lacks real-time environmental interaction, and its cognitive boundaries are strictly locked by the predefined dataset; if the true global optimal combination is omitted during data construction, the DPO policy can never discover it. In contrast, AuTAgent actively explores the environment. It dynamically generates diverse tool combinations during training, executes them with the frozen reasoner, and receives instant feedback via a baseline-subtracted differential reward. This mechanism allows the agent to actively learn via trial and error in the combinatorial action space, optimizing directly for true causal utility (verifiable net performance gains). Consequently, AuTAgent successfully suppresses redundancies and achieves superior reasoning efficiency.
>
> Please note that all evaluations in the table below were conducted using the **Qwen2-Audio-7B-Instruct** backbone:
>
> | Method | MMAU Test-mini (%) | MMAR (%) |
> | :--- | :--- | :--- |
> | Tools w/ Desc. (Zero-shot) | 51.20 | 37.50 |
> | SFT Paradigm | 52.70 | 39.20 |
> | DPO Paradigm | 53.20 | 39.40 |
> | **AuTAgent (Ours)** | **54.20** | **41.30** |
>
> *(Detailed experimental results are provided in the link: https://anonymous.4open.science/r/AuTAgent1/Reviewer_Udqf_detailed_results.md).*
>
> **Response to Weakness 3: Data Sampling Strategy & Scale**
>
> We apologize for the phrasing "random subsampling" in Sec 4.1; it was actually a strategic sampling designed to maximize learning efficiency. We will correct this in the revision.
>
> 1. **Strategic Domain Distribution:** We sampled AVQA to strictly maintain a 2:1:1 ratio for environmental sounds, speech, and music. Environmental sounds require a larger proportion due to their infinite, open-set nature with chaotic overlapping events, unlike rule-constrained, structured speech and music.
> 2. **The Scaling Trade-off in GRPO:** Unlike SFT, RL requires extensive trial-and-error interactions per sample. In massive-scale training (e.g., 38,000 samples) under a realistic compute budget, the agent lacks sufficient interaction to explore the action space per sample, leading to severe underfitting. Conversely, too little data (1,000 samples) hinders generalization. Our 2,025-sample subset provides the perfect balance: optimal diversity for learning robust policies while ensuring GRPO converges efficiently.
>
> | Training Data Scale | MMAU Test-mini (%) | MMAR (%) | Training Steps |
> | :--- | :--- | :--- | :--- |
> | 1,000 Samples | 51.10 | 38.50 | ~120 |
> | **2,025 Samples (Ours)** | **54.20** | **41.30** | **~200** |
> | 10,000 Samples | 52.60 | 39.70 | ~850 |
> | 38,000 (Full) | 53.30 | 39.10 | >2000 |
>
> *(More detailed charts regarding scaling are available in the link: https://anonymous.4open.science/r/AuTAgent1/Reviewer_Udqf_detailed_results.md).*

---

> > ### Author Rebuttal · Reviewer_Udqf · 2026-04-01
> >
> > Thank you for your response. I understand your answers to the last two points. However, I still have concerns regarding the first point (the similarity to VisTA).
> >
> > While the response elaborates on the inherent differences between audio and visual modalities, as well as between audio and visual tools, these arguments primarily highlight the difficulty of the audio domain rather than the contribution of the tool selection algorithm itself.
> > The core of my concern remains:
> > - What are the key algorithmic departures that distinguish AuTAgent from existing visual tool-use agents?
> > - Given that audio processing presents unique challenges, have the authors introduced specific designs to address these challenges?
> >
> > I would appreciate a clearer explanation of the technical differences between AuTAgent and VisTA.

---

> > > ### Author Response · Authors · 2026-04-01
> > >
> > > We sincerely thank you for your continued engagement and feedback. We fully understand the expectation to see pure, foundational algorithmic innovations in RL. However, we respectfully emphasize that AuTAgent was not conceived to propose a general-purpose RL foundational algorithm, but rather as a pioneering framework designed to resolve the **"epistemological bottlenecks"** and **"tool redundancy"** issues inherent in audio reasoning.
> > >
> > > **1. Capability Expansion (VisTA) vs. Retrieving Lost Perceptual Dimensions (AuTAgent)**
> > >
> > > The core difference lies in the fundamentally distinct epistemological roles tools play within these two domains. VisTA targets the visual domain, where the Visual Large Language Model (VLM) has already received **complete image pixel information**; tools (e.g., Chart-to-table) serve to provide further structural parsing and logical assistance for existing visual inputs.
> > >
> > > In contrast, current Large Audio Language Models (LALMs) suffer from the **Representation Bottleneck**. During the compression process, standard acoustic encoders directly discard fine-grained low-level features (such as exact pitch, frequency, and other physical metrics) to prioritize global semantic information. Therefore, the tools invoked by AuTAgent are not simple "auxiliary analysis tools," but are used to acquire low-level acoustic features that the model **physically cannot perceive**. Without the explicit injection of these external features, the model can only hallucinate based on text priors. VisTA expands the inherent reasoning capabilities of the model, whereas AuTAgent uses external tools to **retrieve the model's lost perceptual dimensions**. This makes our RL framework a necessary component for achieving verifiable acoustic reasoning, rather than merely an add-on feature.
> > >
> > > **2. The Necessity of the RL Exploration Mechanism Over Static Priors**
> > >
> > > In the visual domain, the prior knowledge of large models is typically sufficient to trigger the correct tool (e.g., an input chart naturally corresponds to a chart parsing tool). However, audio is an **opaque, one-dimensional time series**. Models lack the inherent priors to intuitively determine whether a complex, overlapping audio event requires Pitch Tracking, Tempo Estimation, or Sound Classification.
> > >
> > > We rigorously proved this through comprehensive baseline experiments (No-Tool, Random Selection, Tools w/ Desc.). Because LALMs struggle to accurately judge acoustic demands based solely on static text descriptions, both zero-shot and static prompting strategies performed poorly. AuTAgent validates the indispensability of the RL exploration mechanism for the audio domain: it allows the agent to **autonomously discover effective, dynamic tool combinations** through environmental interaction, thereby transcending brittle text-based heuristic rules.
> > >
> > > **3. Clarifying the Scope of Innovation: The Differential Reward Mechanism vs. VisTA's Reward**
> > >
> > > We wish to clarify that our work does not aim to propose a general-purpose RL base algorithm (e.g., DAPO or GSPO). Instead, our algorithmic innovation is a reward design specifically tailored to address **Tool Redundancy** in the audio domain. While VisTA also uses outcome-based rewards, there is a **fundamental difference in handling "zero-gain" tool invocations**.
> > >
> > > In VisTA's reward mechanism, if both the base model and the tool-augmented path answer correctly, it still **assigns a +1**. Furthermore, when tools actively degrade a correct baseline prediction, VisTA applies only a mild **-0.5 penalty**. In the visual domain, this is reasonable since moderate redundant tool usage rarely impacts the final judgment. However, applying this unpenalized mechanism directly to audio tends to **induce "shortcut learning."**
> > >
> > > To mitigate this, our **Baseline-Subtracted Differential Reward** introduces a stricter validation strategy: if the frozen base model answers correctly without tools, the system assigns a **Zero Reward**. Tailored to audio feature density, this explicitly removes rewards for zero-gain operations. It drives the agent toward parsimony, ensuring tools are triggered only for a clear **"net performance gain."** As empirically validated in Table 4, this specific design successfully suppresses redundant tool invocations (1.30 vs. 1.62) while achieving superior reasoning accuracy compared to standard binary rewards, explicitly confirming its algorithmic necessity for the audio domain.
> > >
> > > We sincerely appreciate your time and effort in reviewing our paper. Following your suggestions, we supplemented our manuscript with extensive additional experiments, which profoundly elevated the overall quality of our work.
> > >
> > > We hope this clarifies the unique epistemological value and algorithmic necessity of our approach, and we would be deeply grateful if you might consider re-evaluating our submission in light of these improvements.

---

### Decision · Program_Chairs · 2026-04-30

**Decision:**

Accept (regular)

**Comment:**

The paper received two Weak Accepts and one Weak Reject, with the latter reviewer raising their score from 2 to 3 after the rebuttal. Overall, all reviewers agree that the authors have addressed most of the concerns raised during the initial review phase. The rebuttal was effective in clarifying key issues and strengthening the submission. Based on the reviewers’ consensus and the improvements made, the current decision is Accept. The authors are encouraged to incorporate all relevant rebuttal content into the final revision, especially the points raised by Reviewer Udqf, to further strengthen the clarity and completeness of the paper.